# Ultrasoft Classical Systems at Zero Temperature

**DOI:** 10.3390/e25020356

**Published:** 2023-02-15

**Authors:** Matheus de Mello, Rogelio Díaz-Méndez, Alejandro Mendoza-Coto

**Affiliations:** 1Department of Physics, Kyoto University, Kyoto 606-8502, Japan; 2Departamento de Física, Universidade Federal de Santa Catarina, Florianópolis 88040-900, Brazil; 3Ericsson BA Cloud Software, R&D DSS, Ericsson Building 8, 16440 Kista, Sweden; 4Max-Planck-Institut für Physik Komplexer Systeme, Nöthnitzer Str. 38, 01187 Dresden, Germany

**Keywords:** soft-core interaction potential, GEM-α model, classical ground state

## Abstract

At low temperatures, classical ultrasoft particle systems develop interesting phases via the self-assembly of particle clusters. In this study, we reach analytical expressions for the energy and the density interval of the coexistence regions for general ultrasoft pairwise potentials at zero temperatures. We use an expansion in the inverse of the number of particles per cluster for an accurate determination of the different quantities of interest. Differently from previous works, we study the ground state of such models, in two and three dimensions, considering an integer cluster occupancy number. The resulting expressions were successfully tested in the small and large density regimes for the Generalized Exponential Model α, varying the value of the exponent.

## 1. Introduction

Bounded repulsive interaction potentials whose Fourier transform has a negative minimum are known to be responsible for the self-assembly of cluster structures in systems of fully penetrable particles [1,2,3,4]. In the last decades, these so-called ultrasoft systems have emerged as a preferential model to study interesting collective phenomena that belong to apparently distant domains, from cold atoms [5,6,7,8,9], vortex matter [10,11,12] and nuclear matter [13,14,15] to colloidal and polymeric systems [16,17,18,19,20]. In this way, the interest in describing the general properties of ultrasoft models and their connections to specific hamiltonians includes both theoretical and practical motivations.

While the specific description of self-assembly processes is a general open subject in complex systems science, cluster-forming experimental setups have risen up in recent years within soft and condensed matter [21,22,23] and have been successfully realized in purely repulsive DNA-based systems [24]. In all cases, the cluster formation occurring below the 2d-solid melting temperature in ultrasoft systems closely resembles the properties and topological signatures of the corresponding ground-state phases. It is then straightforward that a full understanding of the zero-temperature diagram of the cluster phases can boost the general comprehension of this emergent phenomenology.

In two and three dimensions, with increasing density, ultrasoft systems undergo an infinite sequence of transitions between cluster-crystalline states with increasing occupancy numbers [2,25,26]. That is, between crystal structures in which the nodes consist of particle clusters of different sizes or occupancies, i.e., number of particles per cluster. Interestingly, these isostructural transitions occur without changing the underlying crystalline symmetries and only increasing the cluster occupancy through the phase separation mechanism expected for a first-order transition. It is worth mentioning that, according to [26], these kinds of systems at finite temperatures develop a cascade of isostructural transitions between cluster states with different occupation numbers that ends in a sequence of critical points.

The described behavior for cluster-forming systems has been well-documented via phase diagrams obtained with computer simulations in a number of ultrasoft systems [4,27,28,29]. From these numerical studies, it is clear that phases with an integer cluster occupation number, i.e., phases in which all clusters have exactly the same number of particles, emerge within intervals instead of at sharp values of density. However, this interplay between phases with integer and fractional occupation, which is key for the accurate description of the low-temperature phase diagrams, has been elusive in many theoretical calculations. Here, it is worth mentioning the work from Likos et al. [26], in which the ground-state properties of the face-centered cubic (FCC) cluster structure for the GEM4 model in three dimensions is properly calculated using Maxwell’s construction.

In this work, we use the method followed in [26] to obtain analytical expressions for the coexistence regions of soft-core pair interaction potentials. These expressions are given as expansions in the inverse of the occupation of the corresponding cluster phase undergoing the isostructural transition as the density is increased. We showed that the obtained analytical expressions are able to accurately describe the phase diagram of the soft-core GEM-α models, even at the transitions occurring at the lowest densities.

In these cases, the clustering mechanism is a way of minimizing the repulsion between particles, and consequently, the system arranges itself into a triangular lattice of particle clusters in two dimensions and in an FCC structure for three dimensions. By focusing on states of clusters with constant integer occupancy, referred to as pure phases, our formalism determines the energetic characteristics of these structures to explore their stability as a function of the density. First-order transitions from a cluster-crystal phase with occupancy *n* to another of n+1 occur through a coexistence region that can be characterized by a fractional occupancy. Therefore, using purely thermodynamical principles, a very general result can be obtained describing the emergence of both pure and mixed phases.

As a significant proof, the method is applied to the generalized exponential model GEM-α, which is a well-known cluster-forming interaction whose low-temperature phase diagram has been explored with numerical simulations for some values of α. We found a perfect agreement between our outcomes and the reported simulation results [4,27]. The additional finding of closed analytical expressions for the emergence of the different phases in the high-density regime also enlarges the relevance of the present study for the soft matter community.

## 2. Analytical Description

The interaction energy of a classical system of particles is given by
(1)$$E=\frac{1}{2}\sum_{i\neq j}V\left({\vec{r}}_{i}-{\vec{r}}_{j}\right).$$
We separately analyze the sequence of transitions occurring for the triangular lattice of clusters in two dimensions and for the FCC cluster lattice in three dimensions. In the following subsections, we present a full characterization of the zero-temperature properties of the system assuming that clusters are formed by superimposed particles.

### 2.1. Triangular Cluster-Crystal in Two Dimensions

The energy per particle of a two-dimensional triangular lattice of *n*-particle clusters, with lattice spacing $a_{n}$, is given by
(2)$$\frac{E}{N}=\frac{1}{2}\left(n\sum_{p,q}V(|{\vec{r}}_{p,q}\left|\right)-V\left(0\right)\right),$$
where ${\vec{r}}_{p,q}=a_{n}\left(p{\vec{e}}_{1}+q{\vec{e}}_{2}\right)$, *p* and *q* are integers, and the basis vectors representing the triangular lattice are taken as ${\vec{e}}_{1}=\left(1,0\right)$ and ${\vec{e}}_{2}=\left(-1/2,\sqrt{3}/2\right)$. Since the system is organized in a triangular lattice of clusters formed by *n* superimposed particles, with lattice parameter $a_{n}$, it is straightforward to conclude that the average density of the system will be
(3)$$\rho =\frac{2n}{\sqrt{3}a_{n}^{2}}.$$
This relation allows us to calculate the lattice parameter $a_{n}\left(\rho \right)$, at a given density, for configurations with any cluster occupation. The expression (Equation 2) can still be written in a more interesting way by rewriting it in terms of the Fourier transform of the interaction potential $\hat{V}\left(\vec{k}\right)$, defined as
(4)$$\hat{V}\left(k\right)=\int \frac{d^{2}r}{{\left(2\pi \right)}^{2}}e^{i\vec{k}\cdot \vec{r}}V\left(r\right).$$

Now, we can take advantage of the identity
(5)$$\sum_{p,q}V(|{\vec{r}}_{p,q}\left|\right)=\frac{2}{\sqrt{3}a_{n}^{2}}\sum_{p,q}\hat{V}\left(\right|{\vec{k}}_{p,q}\left|\right),$$
where the set of wave vectors ${\vec{k}}_{p,q}=k_{0}\left(n,\rho \right)\left(pe_{1}^{\prime }+qe_{2}^{\prime }\right)$ represent the reciprocal lattice vectors of the corresponding triangular lattice ${\vec{r}}_{p,q}$. Considering our previous choice of ${\vec{r}}_{p,q}$, the basis vectors of the set ${\vec{k}}_{p,q}$ can be taken as $e_{1}^{\prime }=\left(0,1\right)$ and $e_{2}^{\prime }=\left(\sqrt{3}/2,-1/2\right)$, and the lattice size is given by $k_{0}\left(n,\rho \right)=4\pi /(\sqrt{3}a_{n}\left(\rho \right))$. In Appendix A, we present a demonstration of the identity of Equation (Equation 5), which allows rewriting of the energy per particle of the triangular lattice as
(6)$$\frac{E}{N}=E_{n}=\frac{1}{2}\left(\rho \sum_{p,q}\hat{V}\left(\right|{\vec{k}}_{p,q}\left|\right)-V\left(0\right)\right).$$
The above expression corresponds to the exact energy of a pure triangular cluster lattice with a given integer cluster occupation number and density. As we will see, this information is sufficient to calculate all the ground-state properties, including the classical ground-state phase diagram. The first building block to calculate the properties of the sequence of phase transitions, occurring as density is increased, will be the study of the crossing energies between energy curves corresponding to clusters with integer consecutive occupation numbers. In the next section, we focus on the analytical treatment of this problem.

#### 2.1.1. Energy Crossing Densities

In general, each first-order transition occurring as the density is increased is given by a crossover region in which the occupancy number grows continuously from a given integer value *n* to its consecutive n+1. A first estimation of the location of these phase transitions can be obtained by calculating the densities at which the energy curves corresponding to each consecutive integer occupation number cross. Imposing the condition $E_{n}\left(\rho_{n}\right)=E_{n+1}\left(\rho_{n}\right)$, we reach the condition
(7)$$f_{V}\left(k_{0}\left(n,\rho_{n}\right)\right)=f_{V}\left(k_{0}\left(n+1,\rho_{n}\right)\right),$$
where
(8)$$f_{V}\left(k\right)=\sum_{p,q}\hat{V}\left(k\right|pe_{1}^{\prime }+qe_{2}^{\prime }\left|\right).$$

Since $k_{0}\left(n,\rho \right)/k_{0}\left(n+1,\rho \right)=\sqrt{(n+1)/n}$, we know that in the limit n→∞, $k_{0}\left(\rho ,n\right)\to k_{0}\left(n+1,\rho \right)$. This implies that, if Equation (Equation 7) has a sequence of solutions consistent with repeated transitions from clusters with occupation *n* to n+1, the function $f_{V}\left(k\right)$ must have a local maximum or minimum at some value $k_{m}$, around which the sequence of values of $k_{0}\left(n,\rho_{n}\right)$ and $k_{0}\left(n+1,\rho_{n}\right)$ at the transition, converges to $k_{m}$ as n→∞. Considering that in our case an increase in the density always produces transitions in which the cluster occupation number increases from *n* to n+1, it can be concluded that $f_{V}\left(k\right)$ must have a local minimum at $k_{m}$. Consequently, around $k=k_{m}$, $f_{V}\left(k\right)$ can be approximated by a quadratic form of the type
(9)$$f_{V}\left(k\right)=f_{V}\left(k_{m}\right)+a/2{\left(k-k_{m}\right)}^{2}.$$

From Equation (Equation 7), and considering that for large enough *n* the values of $k_{0}\left(n,\rho_{n}\right)$ and $k_{0}\left(n+1,\rho_{n}\right)$ at the transition are close to $k_{m}$, we can reach the condition $k_{0}\left(n,\rho_{n}\right)+k_{0}\left(n+1,\rho_{n}\right)=2k_{m}$. This condition allows us to estimate the density $\rho_{n}$ at which the energies of the *n* and n+1 phases coincide, which yields
(10)$$\rho_{n}=\frac{\sqrt{3}k_{m}^{2}n\left(n+1\right)}{2\pi^{2}{\left(\sqrt{n+1}+\sqrt{n}\right)}^{2}}.$$
This result is expected to be valid in the large *n* limit. Nevertheless, several cases were tested, producing relatively good estimates, even for n=1.

Regarding the dependence of the ground-state energy on the density, in general, we already know that
(11)$$E_{n}\left(\rho \right)=\frac{1}{2}\left(\rho f_{V}\left(k_{0}\left(n,\rho \right)\right)-V\left(0\right)\right),$$
which is valid for densities ρ in the interval $\left(\rho_{n-1},\rho_{n}\right)$. In the asymptotic regime n≫1, Equation (Equation 11) can be approximated by
(12)$$E_{n}\left(\rho \right)=\frac{1}{2}\left(\rho \left(f_{V}\left(k_{m}\right)+\frac{a}{2}{\left(k_{0}\left(n,\rho \right)-k_{m}\right)}^{2}\right)-V\left(0\right)\right).$$
The family of energy functions defined by Equation (Equation 11) describes the ground-state energy of the system considering only pure phases, i.e., without considering the existence of coexistence regions.

It is worth comparing these results with the classical mean-field results, in which *n* is taken as a variational parameter. In the latter case, it is straightforward to conclude that *n* varies continuously with density, in such a way that $k_{0}\left(n,\rho \right)=k_{m}$. This means that in the usual mean-field approximation
(13)$$E_{MF}\left(\rho \right)=\frac{1}{2}\left(\rho f_{V}\left(k_{m}\right)-V\left(0\right)\right).$$
Interestingly, this curve represents the envelope curve of the family of curves $E_{n}\left(\rho \right)$ given by Equation (Equation 12). Moreover, at low enough densities, we can see that $E_{MF}\left(\rho \right)<0$, which is clearly a drawback of the mean-field approach, since for purely repulsive potentials the ground-state energy will always be a positive definite quantity. Finally, we would like to add that it is in principle possible to improve the mean-field description using more sophisticated calculations schemes in order to properly capture the low-temperature and density regimes of soft-core particles systems, as shown by Prestipino et al. in ref. [30].

#### 2.1.2. Coexistence Regions

As mentioned above, the first-order transitions between different cluster-crystal phases, as the density is increased, occur through a coexistence region in which the occupancy number has a crossover from *n* to n+1. The densities corresponding to the beginning and the end of the coexistence regions can be determined by means of thermodynamic principles. Within the coexistence regions, the pressure P(ρ) and the chemical potential μ(ρ) of each pure cluster phase are equal and remain constant, while the density is increased along the whole coexistence region. The mathematical condition determining the densities $\rho_{1n}$ and $\rho_{2n}$ at the beginning and the end of the coexistence region corresponding to the transition of a cluster with occupancy number *n* to n+1 is given by
(14)$$\begin{array}{ccc} P_{n}\left(\rho_{1n}\right) & = & P_{n+1}\left(\rho_{2n}\right)\\ \mu_{n}\left(\rho_{1n}\right) & = & \mu_{n+1}\left(\rho_{2n}\right). \end{array}$$
These pressures and chemical potentials can be calculated using the relations $P_{n}\left(\rho \right)=\rho^{2}\frac{\partial E_{n}\left(\rho \right)}{\partial \rho }$ and $\mu_{n}\left(\rho \right)=E_{n}\left(\rho \right)+\rho \frac{\partial E_{n}\left(\rho \right)}{\partial \rho }$.

The system of Equation (Equation 14) cannot be solved in general for an arbitrary potential, even considering the large *n* limit of the functions $E_{n}\left(\rho \right)$ given in Equation (Equation 12). To proceed, we look for a solution for $\rho_{1n}$ and $\rho_{2n}$ in the form of a large *n* expansion of the type
(15)$$\begin{array}{ccc} \rho_{1n} & = & a_{0}n+a_{1}+\frac{a_{2}}{n}+\ldots \\ \rho_{2n} & = & b_{0}n+b_{1}+\frac{b_{2}}{n}+\ldots \;, \end{array}$$
and solve the set of Equation (Equation 14) order by order. It is worth noticing here that, due to the nature of the power expansion in Equation (Equation 15), the coefficients $\left\{a_{j}\right\}$ and $\left\{b_{j}\right\}$ are *n*—independent. However, this does not affect the convergence of $\rho_{1n}$ and $\rho_{2n}$; in any case, we realize that as long as $\sum_{j}a_{j}$ and $\sum_{j}b_{j}$ converge, the expansions will be well-defined. A simple argument that can be raised in general about the convergence of such expansions is that, by construction, they are the solution of the system of Equation (Equation 14), so as long as a coexistence region exists, the convergence is assured.

We should note that since the r.h.s. of Equation (Equation 10) grows linearly with *n*, the expansion’s leading term for $\rho_{1n}$ and $\rho_{2n}$ should thus be O(*n*). Additionally, as we will see, the subleading terms of order O(1) and $O(\frac{1}{n})$ significantly improve the accuracy of the proposed expansions.

Considering the perturbative structure of $\rho_{1n}$ and $\rho_{2n}$, we can conclude that to calculate the coexistence regions up to $O(\frac{1}{n})$, we must keep terms up to $O({\left(k-k_{m}\right)}^{3})$ in $f_{V}\left(k\right)$. The expansion of $f_{V}\left(k\right)$ up to the third order reads
(16)$$f_{V}\left(k\right)\approx f_{0}+\frac{f_{2}}{2!}{\left(k-k_{m}\right)}^{2}+\frac{f_{3}}{3!}{\left(k-k_{m}\right)}^{3}+...,$$
where $f_{n}=f_{V}^{\left(n\right)}\left(k_{m}\right)$ represents the *n*-derivative of the function $f_{V}\left(k\right)$ evaluated in $k=k_{m}$. Solving perturbatively the system of Equation (Equation 14) in powers of *n*, we obtain
(17)$$\begin{array}{ccc} a_{0} & = & b_{0}=\frac{\sqrt{3}k_{m}^{2}}{8\pi^{2}}\\ a_{1} & = & \frac{\sqrt{3}k_{m}^{2}f_{0}}{2\pi^{2}\left(f_{2}k_{m}^{2}+8f_{0}\right)}\\ b_{1} & = & \frac{\sqrt{3}k_{m}^{2}\left(f_{2}k_{m}^{2}+4f_{0}\right)}{8\pi^{2}\left(f_{2}k_{m}^{2}+8f_{0}\right)}\\ a_{2} & = & b_{2}=-\frac{k_{m}^{2}f_{0}^{2}\left(9f_{2}+f_{3}k_{m}\right)\left(3f_{2}k_{m}^{2}+8f_{0}\right)}{2\sqrt{3}\pi^{2}f_{2}{\left(f_{2}k_{m}^{2}+8f_{0}\right)}^{3}}. \end{array}$$
The obtained result for the coexistence boundaries of the first-order transition from a cluster with occupancy *n* to n+1, given in Equations (Equation 15) and (Equation 17), was compared with the numerical solutions of Equation (Equation 14) for some specific models, and the agreement, even for n=1, is surprisingly good. Details of this comparison can be found in Section 3.1.1.

Finally, we turn to the question of how to determine the energy of the ground state within the coexistence region. To exactly calculate the behavior of the energy in this regime, we take advantage of the fact that, within this region, the pressure of the system remains constant and equal to the coexistence pressure $P_{c}$. Integrating the equation defining pressure in the system of Equation (Equation 14), considering $P_{c}$ as a constant, we obtain that, within the coexistent region,
(18)$$E_{n}\left(\rho \right)=E_{n}\left(\rho_{1n}\right)+P_{c}\left(\frac{1}{\rho_{1n}}-\frac{1}{\rho }\right).$$
This result is completely general and valid in the two-dimensional, as well as in the three-dimensional, case.

### 2.2. FCC Cluster-Crystal in Three Dimensions

The procedures followed in the previous section to study the ground state of cluster-forming systems in two dimensions can be generalized to the three-dimensional case without major difficulties. Numerical simulations, as well as direct calculations, allow concluding that, among all possible three-dimensional crystals, the one minimizing the energy of the system under consideration is the FCC lattice. This is actually not surprising since the FCC is one of the closest packed structures in three dimensions.

As before, the energy of a cluster-crystal of occupation with *n* particles per lattice site is given by
(19)$$\frac{E}{N}=\frac{1}{2}\left(n\sum_{p,q,s}V(|{\vec{r}}_{p,q,s}\left|\right)-V\left(0\right)\right),$$
where ${\vec{r}}_{p,q,s}$ represents the position of the clusters in an FCC structure. We choose ${\vec{r}}_{p,q,s}=a_{n}(p{\vec{v}}_{1}+q{\vec{v}}_{2}+s{\vec{v}}_{3}$), where the basis vectors are taken as ${\vec{v}}_{1}=\left(0,1,1\right)1/2$, ${\vec{v}}_{2}=\left(1,0,1\right)1/2$ and ${\vec{v}}_{3}=\left(1,1,0\right)1/2$.

For an FCC lattice of clusters, with *n* particles per site, the average density is given by
(20)$$\rho =\frac{4n}{a_{n}^{3}},$$
where $a_{n}$ represents the lattice spacing of the structure. This relation allows us to calculate the lattice spacing in terms of the particle occupation of the clusters and the average density.

Following the same method described in the two-dimensional case, we can rewrite the energy per particle of the system in the form
(21)$$\frac{E}{N}=E_{n}=\frac{1}{2}\left(\rho \sum_{p,q,s}\hat{V}\left(\right|{\vec{k}}_{p,q,s}\left|\right)-V\left(0\right)\right),$$
where the wave vectors in the sum are taken as ${\vec{k}}_{p,q,s}=\left(2\pi \sqrt{3}/a_{n}\right)\left(pv_{1}^{\prime }+qv_{2}^{\prime }+sv_{3}^{\prime }\right)$, and the basis vectors are given by $v_{1}^{\prime }=\left(1,1,-1\right)1/\sqrt{3}$, $v_{2}^{\prime }=\left(1,-1,1\right)1/\sqrt{3}$ and $v_{3}^{\prime }=\left(-1,1,1\right)1/\sqrt{3}$.

Now, we can find the densities at which the pure phases change stability. At these densities, the condition $E_{n}\left(\rho \right)=E_{n+1}\left(\rho \right)$ implies that
(22)$$g_{V}\left(k_{0}\left(n,\rho_{n}\right)\right)=g_{V}\left(k_{0}\left(n+1,\rho_{n}\right)\right),$$
where $g_{V}\left(k\right)=\sum_{p,q,s}\hat{V}\left(k\right|pv_{1}^{\prime }+qv_{2}^{\prime }+sv_{3}^{\prime }\left|\right)$ and $k_{0}\left(n,\rho_{n}\right)=2\pi \sqrt{3}/a_{n}\left(\rho \right)$, where $a_{n}\left(\rho \right)$ represents the lattice spacing of the FCC lattice related to the average particle density by Equation (Equation 20). In this case, as in the two-dimensional case, when n≫1, $a_{n}\left(\rho \right)/a_{n+1}\left(\rho \right)\to 1$. This means that, once again, in order to have a sequence of cluster transitions with increasing density, $g_{V}\left(k\right)$ needs to have a minimum at some finite value $k_{m}$. Consequently, for values of *k* close enough to $k_{m}$, we can approximate $g_{V}\left(k\right)$ by its expansion up to the second order
(23)$$g_{V}\left(k\right)=g_{V}\left(k_{m}\right)+\frac{a}{2}{\left(k-k_{m}\right)}^{2}.$$

The condition given in Equation (Equation 22) leads again to the conclusion that $k_{0}\left(n,\rho_{n}\right)+k_{0}\left(n+1,\rho_{n}\right)=2k_{m}$. This equation allows estimating the density at the cluster transition in the large cluster occupation limit, considering only pure phases
(24)$$\rho_{n}=\frac{4k_{m}^{3}n\left(1+n\right)}{3\sqrt{3}\pi^{3}{\left(n^{1/3}+{\left(1+n\right)}^{1/3}\right)}^{3}}.$$

Analogous to the two-dimensional case, the system of Equation (Equation 14) can now be solved for the FCC cluster crystal in order to find the densities limiting the coexistence region for each first-order transition. Expanding $g_{V}\left(k\right)$ up to the third order, we obtain
(25)$$g_{V}\left(k\right)\approx g_{0}+\frac{g_{2}}{2!}{\left(k-k_{m}\right)}^{2}+\frac{g_{3}}{3!}{\left(k-k_{m}\right)}^{3}+...,$$
where $g_{n}=g_{V}^{\left(n\right)}\left(k_{m}\right)$ represents the *n*-derivative of the function $g_{V}\left(k\right)$ evaluated in $k=k_{m}$.

The system of Equation (Equation 14) is solved now perturbatively in powers of $n^{-1}$ order by order, considering that the densities defining the boundaries of the coexistence region, for each first-order transition, have the form given in Equation (Equation 15). This solution process leads to the following coefficients for the three-dimensional case:(26)$$\begin{array}{ccc} a_{0} & = & b_{0}=\frac{k_{m}^{3}}{6\sqrt{3}\pi^{3}}\\ a_{1} & = & \frac{\sqrt{3}k_{m}^{3}g_{0}}{2\pi^{3}\left(g_{2}k_{m}^{2}+18g_{0}\right)}\\ b_{1} & = & \frac{k_{m}^{3}\left(g_{2}k_{m}^{2}+9g_{0}\right)}{6\sqrt{3}\pi^{3}\left(g_{2}k_{m}^{2}+18g_{0}\right)}\\ a_{2} & = & b_{2}=-\frac{3\sqrt{3}k_{m}^{3}g_{0}^{2}\left(12g_{2}+g_{3}k_{m}\right)\left(g_{2}k_{m}^{2}+6g_{0}\right)}{4\pi^{3}g_{2}{\left(g_{2}k_{m}^{2}+18g_{0}\right)}^{3}}. \end{array}$$

As in the two-dimensional case, the analytical solutions found here showed very good agreement with the exact numerical results, even for low values of the cluster occupancy number *n*. This kind of analytical expression can be also useful to gain insights into the behavior of the cluster crystals at low temperatures.

## 3. Numerical Results with GEM-α

In order to test our analytical approach, the method developed in Section 2 is used to characterize exactly the classical ground-state of the GEM-α model, which is given by a pairwise interaction of the form
(27)$$V\left(r\right)=\exp\left(-r^{\alpha }\right).$$
This effective potential has been commonly used in the literature to model systems of dendrimers, star-shaped polymers and general colloidal and polymeric system [16,17]. The GEM-α model represents a bounded, repulsive interaction, whose Fourier transform has a negative minimum at some wave vector for α>2 in all dimensions. It is therefore an ultrasoft model that presents cluster phases with superimposed particles of the type studied in the present work.

### 3.1. Two-Dimensional Case

To illustrate the general behavior of the GEM-α model for a specific α>2, in this section, we study the exact properties of the GEM-4 model. In Figure 1, we show the behavior of the ground-state energy in a wide range of densities. In this figure, the red curve is formed by the different energy branches $E_{n}\left(\rho \right)$ corresponding to each pure cluster-crystal phase. The intersection between the different branches can be identified by the sharp peaks in the red curve (see the figure inset). At the same time, we understand that the first-order transition between the different cluster phases, as the density is increased, occurs through a coexistence region in which the cluster occupancy number varies from *n* to n+1.

The beginning ($\rho_{1n}$) and the end ($\rho_{2n}$) of each coexistence region can be determined numerically by solving the system of Equation (Equation 14). Solving this system of equations allows for determining not only the boundaries of each coexistence region but also the pressure and the chemical potential within the coexistence region. Additionally, such information can be used to determine the behavior of the total energy per particle within the coexistence region by means of Equation (Equation 18).

The coexistence region for each transition is represented by a shaded region in green. At the top of each shaded area, the exact value of the energy per particle is presented by the green solid curves. In the inset of Figure 1, we present a zoom of the original figure in a smaller density interval containing the first cluster transition, from the single-particle triangular lattice to the two-particle cluster-crystal. In this figure, the differences between the energy curves corresponding to the pure phases and those of the coexistence region are evident. Additionally, the beginning and the end of each coexistence region are highlighted with green dots. As we can observe, the presence of the coexistence region results in a further minimization of the ground-state energy. From a thermodynamical point of view, this is precisely why coexistence appears in a first-order transition: it is a mechanism of free energy minimization.

Once we have the exact ground-state energy curve, we can calculate the corresponding pressure and chemical potentials by means of the definitions given in Equation (Equation 14). In Figure 2, we present the exact behavior of the pressure (panel A) and the chemical potential (panel B). The red curves are associated with the behavior within pure cluster-crystal phases in which the occupancy number takes integer values. For states within the coexistence regions, the curves of pressure and chemical potential remain constant, as expected from thermodynamic principles. Once again, the boundaries of the coexistence regions are highlighted with green points.

Now that the general properties of the ground state of the GEM-4 model have been described, we take one step further in the systematic characterization of the GEM-α model. In Figure 3, we present the exact ground-state phase diagram varying the exponent α of the interaction potential and the particle density of the system. The shaded regions in green represent the coexistence regions associated with each first-order transition between different cluster states. At the same time, the densities at which the pure phases change stability, as the density is increased, are represented by red dashed lines. As expected, these lines are always found within the coexistence region corresponding to each first-order phase transition.

It can be noted that, as the value of α→2, the position of the phase transitions moves progressively to infinity, while the coexistence regions shrink to zero. This behavior can be understood considering the properties of the function $f_{V}\left(k\right)$ in the limit α→2. In this regime, $k_{m}\left(\alpha \right)\to \infty$ and $f_{2}\left(k_{m},\alpha \right)\to 0$; this implies, according to Equation (Equation 17), that $a_{0}\to \infty$ and $a_{1}\to b_{1}$, which explains the observed behavior of the coexistence regions.

#### 3.1.1. Comparison between Analytical and Numerical Results

In this section, we compare some of our analytical predictions with their numeral counterparts for the GEM-α model in the two-dimensional case. For the purpose of comparison, we reobtain the expression of the density $\rho_{n}$ at the crossing of the branches $E_{n}\left(\rho \right)$ and $E_{n+1}\left(\rho \right)$ in the two-dimensional case. As mentioned before, during the calculation of $\rho_{1n}$ and $\rho_{2n}$, to obtain expressions with accuracy $O(\frac{1}{n})$, we need to consider an expansion of $f_{V}\left(k\right)$ up to $O({\left(k-k_{m}\right)}^{3})$. Thus, although Equation (Equation 10) is the exact solution considering $f_{V}\left(k\right)=f_{V}\left(k_{m}\right)+a/2{\left(k-k_{m}\right)}^{2}$, it is correct for the original problem only up to order $O({\left(\frac{1}{n}\right)}^{0})$.

For improving this result, we consider $f_{V}\left(k\right)$, given as in Equation (Equation 16), and propose an energy crossing density $\rho_{n}$ in the form of a large *n* expansion of the same order of Equation (Equation 15)
(28)$$\rho_{n}=d_{0}n+d_{1}+\frac{d_{2}}{n}.$$
In this case, the perturbative solution of Equation (Equation 7) order by order leads to the coefficients
(29)$$\begin{array}{ccc} d_{0} & = & \frac{\sqrt{3}k_{m}^{2}}{8\pi^{2}}\\ d_{1} & = & \frac{\sqrt{3}k_{m}^{2}}{16\pi^{2}}\\ d_{2} & = & -\frac{k_{m}^{2}\left(9\sqrt{3}f_{2}+\sqrt{3}k_{m}^{2}f_{3}\right)}{384\pi^{2}f_{2}}, \end{array}$$
where the coefficients $f_{n}$ and $k_{m}$ are defined in Equation (Equation 16).

In Figure 4, we show a comparison between the analytical predictions and the exact numerical results for the GEM-α model in two dimensions. We show two different phase transitions: from the simple crystal n=1 to the two-particle cluster-crystal n=2 (panel A), and the transition from the cluster-crystal n=10 to the cluster-crystal n=11 (panel B).

The dots represent the numerical results, and the continuous curves represent the analytical large *n* expressions given by Equations (Equation 15) and (Equation 17) for the coexistence boundaries and Equations (Equation 28) and (Equation 29) for the densities at the energy crossing. Green full curves correspond to the boundaries of the coexistence regions in each case, while red full curves are related to the energy crossing of the two relevant phases involved in the phase transition.

As can be observed, there is a high degree of agreement between the numerical results and the analytical predictions obtained in the large *n* limit already for the n=10 case, within the full range of models considered. On the other hand, for the lowest possible value of the cluster occupancy (n=1), there is still a good agreement between analytical predictions and numerical exact results in the whole range of α considered, especially regarding the description of the boundaries of the coexistence region.

### 3.2. Three-Dimensional Case

For completeness, in this section, we apply the formalism described in Section 2 to study the GEM-α model in three dimensions. For values α≤2, the function $g_{V}\left(k\right)$ does not have a minimum at a finite value $k_{m}$, and therefore, the system orders in a simple, noncluster crystalline state. Nevertheless, as density increases in this regime, a structural transition occurs from an FCC to a BCC structure, accompanied by a very narrow coexistence region. For α>2, there is an infinite sequence of transitions between different FCC cluster-crystal phases in which the occupation number of the clusters increases with density. In this regime, the FCC cluster structure always has lower energy than the BCC cluster arrangement.

In Figure 5, the α versus density phase diagram is constructed for the 3D system, comparing the energy per particle of the different cluster-crystal phases organized in an FCC lattice and in a single-particle BCC lattice. In the α≤2 regime, as mentioned before, there is no cluster formation and, instead, only a first-order structural phase transition takes place from the FCC lattice to the BCC lattice as the density is increased. The coexistence region associated with this transition is very narrow, and consequently, it is barely visible in Figure 5.

In the regime α>2, in which cluster formation occurs, the scenario is similar to the one observed in the two-dimensional case. In Figure 5, the shaded green areas represent the different coexistence regions associated with each cluster transition. The red dashed lines, as before, provide the boundaries at which the different pure cluster-crystal phases change stability. Finally, the white regions of the phase diagram correspond to pure FCC cluster-crystal phases.

## 4. Concluding Remarks

We analyzed the ground-state phase diagram of ultrasoft systems of particles undergoing a sequence of isostructural cluster transitions as the density of particles is increased, in two and three dimensions. We found closed analytical expressions for the boundary of the coexistence regions as expansions in the inverse of the number of particles per cluster. In this way, we were able to construct the ground-state phase diagram for the GEM-α family of pair interaction potentials producing accurate results, even for the transitions taking place at the lowest densities.

If we compare our zero temperature results with related works in the literature, we observe that, although it has been considered that the occupancy number of clusters can be treated as a variational real parameter, this assumption in general produces quite different phase diagrams when compared with the exact behavior. This version of the mean field approximation can be understood as an attempt of describing the cluster transitions at high densities, which neglects the fact that such first-order transition occurs through phase separation and not irregularly distributing particles over the lattice of clusters. In this context, coexistence regions are not usually identified, since pure clusters phases do not exist in extended regions of the phase diagrams. A nice implementation of the mean-field technique free of the above-mentioned problems can be found in ref. [30].

Finally, we would like to add that the assumption of stability of cluster phases with a homogeneous occupancy number allows to conclude the existence of first-order transitions and not crossovers between the different cluster phases. The properties of these first-order transitions were calculated from very general thermodynamic considerations, and consequently, our results show full consistency with what is expected from a physical point of view. All analytical results show excellent agreement with the exact numerical results presented and with previous numerical simulation results obtained for specific GEM-α models [4,27].

## Figures and Tables

**Figure 1 entropy-25-00356-f001:**
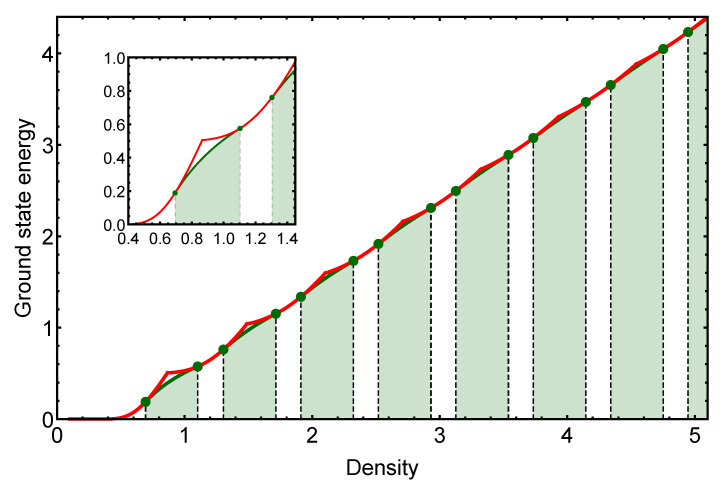
Exact ground-state energy for the GEM-4 model. The red curve corresponds to the ground-state energy of the system considering only pure cluster-crystal phases. The change in stability for each pure cluster phase can be identified by the sharp peaks in the red curve. The coexistence regions corresponding to each cluster-crystal transition are represented by the shaded green areas. In these regions, the actual ground-state energies are represented by the green solid curves. The inset panel is a zoom of the original figure in a density interval corresponding to the energy branches $E_{1}\left(\rho \right)$ and $E_{2}\left(\rho \right)$, given by Equation (Equation 11). It is worth noting how the coexistence mechanism results in further energy minimization when the energy of the mixed phase is compared with the energies of the pure phases involved in the phase transition.

**Figure 2 entropy-25-00356-f002:**
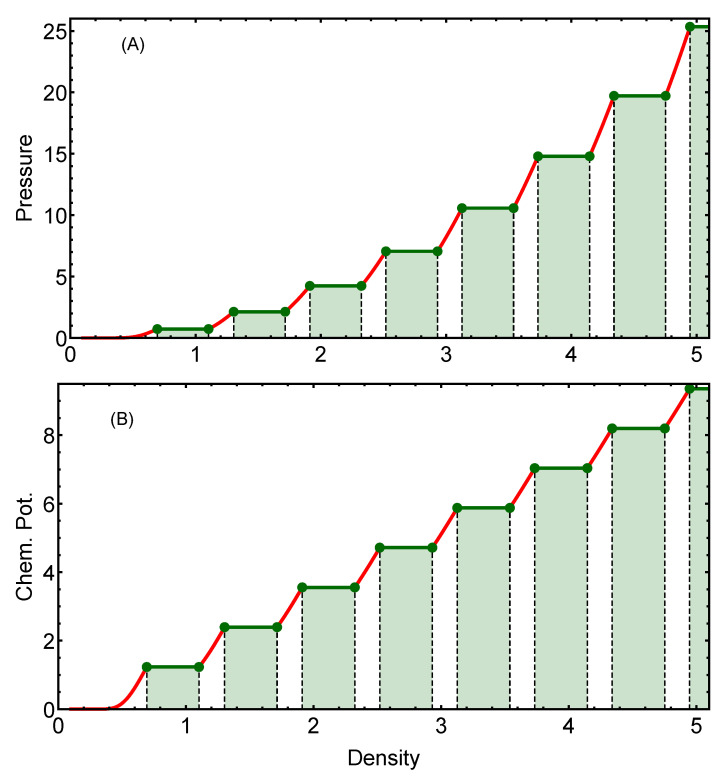
Pressure (**A**) and chemical potential (**B**) as a function of the density for the GEM-4 model in two dimensions. Red curves in both figures correspond to the behavior of the specific magnitude within the pure cluster phases. Green curves correspond to the behavior of the specific magnitude within the coexistence regions. The solid dots define the boundaries of the coexistence regions associated with each first-order phase transition.

**Figure 3 entropy-25-00356-f003:**
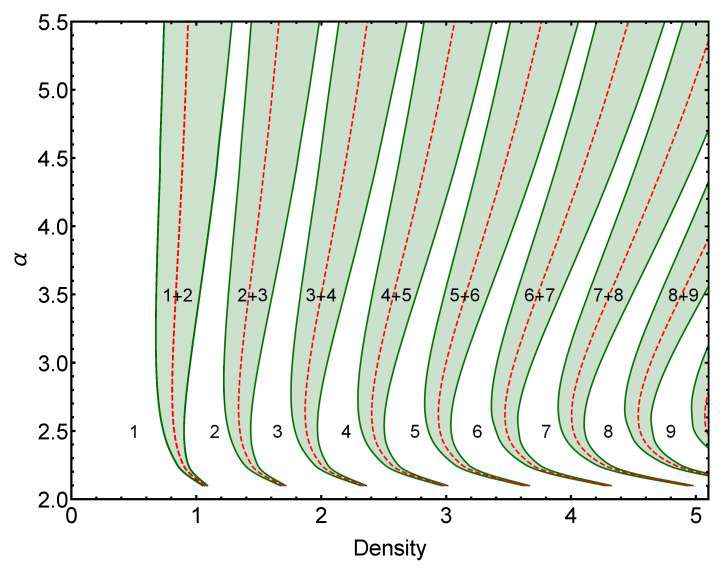
Phase diagram α versus density for the GEM-α model in two dimensions. For all α>2, the system undergoes an infinite sequence of transitions between different cluster-crystal phases as the density is increased. These transitions are accompanied by coexistence regions represented in the figure by the shaded green areas. The dashed red curves represent the densities at which the pure cluster phases change stability.

**Figure 4 entropy-25-00356-f004:**
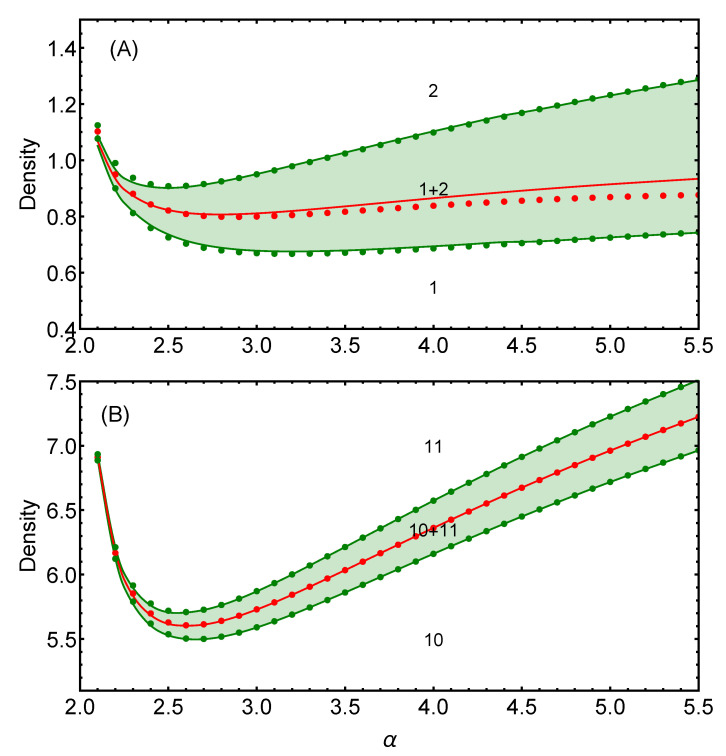
Comparison of the analytical and exact numerical predictions for the densities as a function of the exponent α of the GEM-α model for specific transitions between different cluster states. The red dots represent the numerical exact values of the densities at the crossing of consecutive energy branches, while the continuous red curve represents the analytical asymptotic prediction given by Equations (Equation 28) and (Equation 29). The green dots correspond to the numerical exact values of the boundaries of the coexistence region, while the green continuous lines are given by the analytic predictions in Equations (Equation 15) and (Equation 17). Panel (**A**) corresponds to the transition from triangular lattice n=1 to the two-particle cluster n=2. Panel (**B**) corresponds to the transition from the cluster crystal with (n=10) to the cluster crystal with n=11.

**Figure 5 entropy-25-00356-f005:**
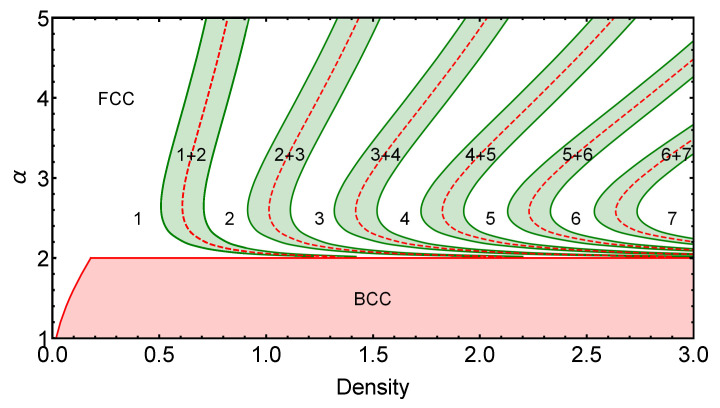
Phase diagram α versus density for the GEM-α model in three dimensions. For values of α≤2, the system orders in a single-particle array presenting a structural transition from an FCC lattice to a BCC lattice, for high enough densities. For α>2, a sequence of different FCC cluster-crystal phases occurs as the density is increased. These phases are represented by the white areas in the figure and have a well-defined number of particles per cluster that increases with density. The dashed red curves represent the densities at which the pure cluster phases change stability, while the green shaded regions represent the coexistence regions associated with each cluster-crystal phase transition.

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
