# Peer review of "Ultrasoft Classical Systems at Zero Temperature"

_entropy, 2023, doi:10.3390/e25020356_

Round 1
Reviewer 1 Report
See attached file

Author Response
Dear referee
Please find in the attachment a point by point answer to all your comments and criticisms.

Reviewer 2 Report
The authors introduce a method for the determination of the ground state (GS) of classical systems of fully penetrable particles interacting via soft-core potentials.
I find the manuscript rather well written (aside from very minor issues) and well structured; however, I have found a very serious issue with the presented results and discussion, which in my eyes must be resolved.
Specifically, the authors present the method they introduce as exact, i.e., approximation-free. However, in the discussion they often make use of simplifications and/or approximations which take place in the limit of cluster size $n >> 1$ (e.g., considering only corrections up to $O(1/n)$ in the analysis of eq. (15)), while saying in some cases that results are "relatively good [...] even for $n = 1$".
Given that the cluster size is a fixed quantity in any calculation (unlike, e.g., the system size, which can be extrapolated to infinity for thermodynamic limit calculations) from the discussion of the authors I gather that their method is in fact not exact for finite cluster sizes, as it relies on an approximation which can be very far from the actual situation in the system (what if I need to work at low densities, i.e., small clusters ?).
I believe that the authors should make the effort to divide the discussion of the method in two sections, one where they do not make any approximations on the cluster size (or any other variables) and the other where such approximations are applied, if deemed necessary. If the technique as discussed in the former section already allows to determine the GS, then the method can indeed be regarded as exact. If this is not the case, however, the method is only really asymptotically exact in the large-cluster-size limit, which is an important distinction (since the approximation does not hold in all realistic calculations).
As a minor point, I think it should be specified in the paper if the coefficients $a_k$ and $b_k$ in eq. (15) are $n$-independent and decreasing/constant with the order, i.e., if the series is guaranteed to converge for $n > 1$.
Only once the authors deal with the issues discussed above, I can reconsider the manuscript for publication in Entropy (hence the, hopefully temporary, No Answer for Scientific Soundness and Overall Merit in my report).
Author Response
Dear referee, please find in the attachment our reply to all your comments and criticism.
Best regards,
The author.

Reviewer 3 Report
The authors present a very nice study of the T = 0 phase diagrams of ulttrasoft, GEM-n models of penetrable particles, elaborating on the existence of both broad regions of integer occupancy and of coexistence regions of occupancy n and n + 1. They employ analytical approaches accompanied by numerical evaluations and they give a lucid presentation of their results, which are useful and relevant to other researchers. I have three critical remarks:
1. At the conclusions, the authors present some criticism on the mean-field theories of arbitrary occupancy. These theories, however, are not wrong: they are simply valid at finite temperatures, which allow for particle hopping between the sites. The coexistence regions are valid for T = 0. This point has been made clear in Ref. [25] by Neuhaus and Likos.
2. It is worth mentioning that in going from T = 0 to finite T, the system undergoes a cascade of isostructural transitions that end at infinitely many critical points, again see Ref. [25].
3. To strengthen the connection with experiments, the recent experimental realization of cluster crystals should be cited:
Emmanuel Stiakakis, Niklas Jung, Natasa Adzic, Taras Balandin, Emmanuel Kentzinger, Ulrich Rücker, Ralf Biehl, Jan K. G. Dhont, Ulrich Jonas, and Christos N. Likos, Self assembling cluster crystals from DNA based dendritic nanostructres, Nature Communications 12, 7167 (2021). DOI: https://doi.org/10.1038/s41467-021-27412-3
Author Response
Dear referee
Please find in the attachment our response to all your comments and criticism.
Best regards,
The authors

Round 2
Reviewer 1 Report
Referee report of paper entropy-2141990 "Ultrasoft classical systems at zero temperature" by M. de Mello, R. Diaz-Mendez, and A. Mendoza-Coto (revised version).
The revised version of the manuscript complies satisfactorily with the points raised in my previous report. Therefore, I recommend it for publication provided the following corrections are made:
1) Abstract, line 4: temperatures ---> temperature
2) Sec. 1, line 34: develops ---> develop
3) Sec. 2.1.1, line 93: calculations schemes ---> calculation schemes
4) Sec. 2.1.1, line 94: Ref. [30] was added by the Authors in compliance with point 8) of my former report. However, the corresponding entry in the "References" section is not correct. It should be Prestipino et al., Phys. Rev. E 92, 022138 (2015).
5) Sec. 2.1.1, line 94: soft-core particles system ---> soft-core particle systems
6) Sec. 4, line 272: same as point 4) above.
Reviewer 3 Report
The authors have addressed the referees' concerns in their revised version. I have one minor comment left: in line 94 of the manuscript, the authors mention a work of Prestipino et al. and they refer to citation [30], which is written by other authors. Please check this and the references once again before publication.